Methods

# Cost-effective DNA methylation profiling by FML-seq

Joseph W Foley ⓘ, Shirley X Zhu, Robert B West

**Current methods for profiling DNA methylation require costly reagents, sequencing, and labor time. We introduce fragmentation at methylated loci and sequencing (FML-seq), a sequencing library protocol that greatly reduces all these costs. Relative to other techniques tested on the same human cell lines, FML-seq produces similar measurements of absolute and differential cytosine methylation at a fraction of the price. FML-seq enables inexpensive, high-throughput experimental designs for large-scale epigenetics research projects.**

## Introduction

DNA methylation is a crucial and well-studied epigenetic mark in many eukaryotes (1). However, DNA methylation profiling remains one of the only areas of genomics where sequencing has not fully displaced microarrays. Each sequencing-based technique has its own impracticalities. Like the microarrays, several common approaches use nucleobase conversion to distinguish cytosine, which converts to uracil, from methylated cytosine, which does not. Whole-genome bisulfite sequencing (WGBS) (2) simply deaminates the entire genome, and deep whole-genome sequencing is then required to count converted versus unconverted bases at every cytosine. Reduced representation bisulfite sequencing (RRBS) (3) reduces the sequencing cost by first using a methylation-insensitive restriction endonuclease to enrich for genome fragments with high CpG content, but this requires a more cumbersome protocol for library preparation. Recent protocols have improved base conversion by replacing the extreme chemical conditions of bisulfite-catalyzed deamination with an enzyme treatment (4).

Base-conversion sequencing produces a dual-channel signal, analogous to single-nucleotide polymorphism profiling: each base is read several times, sometimes converted and sometimes not, and percent methylation is calculated as the proportion of total reads in which that base was not converted. Some alternative methods produce a single-channel signal, analogous to RNA-seq, ChIP-seq, or ATAC-seq. Methylated DNA immunoprecipitation (MeDIP-seq) (5), like ChIP-seq, enriches DNA fragments with methylated cytosine, though it does not directly identify which base

is methylated. Some approaches (MSCC/Methyl-seq/MRE-seq) use a methylation-blocked restriction endonuclease to digest the whole genome, resulting in an inverse single-channel signal of sequence reads accumulating at unmethylated base positions (6, 7, 8). Other approaches conversely use a methylation-dependent restriction endonuclease, producing a forward single-channel signal of sequence reads accumulating at methylated base positions (9, 10, 11, 12, 13, 14); however, these tend to be long protocols with labor-intensive steps such as cutting bands out of an electrophoresis gel.

Here, we introduce fragmentation at methylated loci and sequencing (FML-seq), a method that combines methylation-dependent restriction digestion with an alternative adapter ligation scheme to greatly reduce the time, labor, and cost compared with previous similar protocols. FML-seq fills the gap for rapid, high-throughput, and cost-effective DNA methylation profiling.

## Results

### The FML-seq method

The protocol for FML-seq comprises only three steps (Fig 1A). First, genomic DNA (gDNA) is digested by a methylation-dependent restriction endonuclease that cuts at a certain distance from the 5-methylcytosine or 5-hydroxymethylcytosine in its motif and leaves a short overhang (10). Second, a master mix is added with combined reagents for sticky-end adapter ligation, preparation of the specially designed adapters (15), and indexing PCR. Finally, a single cleanup without size selection is sufficient to purify the library, because the digestion does not produce unusably short fragments (Fig S1) and the adapter design prevents byproducts without gDNA inserts (Fig S2). The resulting library contains unaltered genome sequences alignable by standard pipelines. Each end of a library fragment is derived from a methylation-dependent digestion, so paired-end sequencing detects two methylated cytosine positions per fragment (Fig 1B). FML-seq represents a substantial simplification of previous protocols based on methylation-dependent digestion.

FML-seq uses sequencing adapters with 5' overhangs of 4 random bases (4N) for efficient sticky-end ligation to the corresponding overhangs of unknown bases resulting from digestion by a restriction endonuclease whose motif includes methylated cytosine. This would be expected to result in an overwhelming

Department of Pathology, Stanford University School of Medicine, Stanford, CA, USA

Correspondence: jwfoley@nus.edu.sg

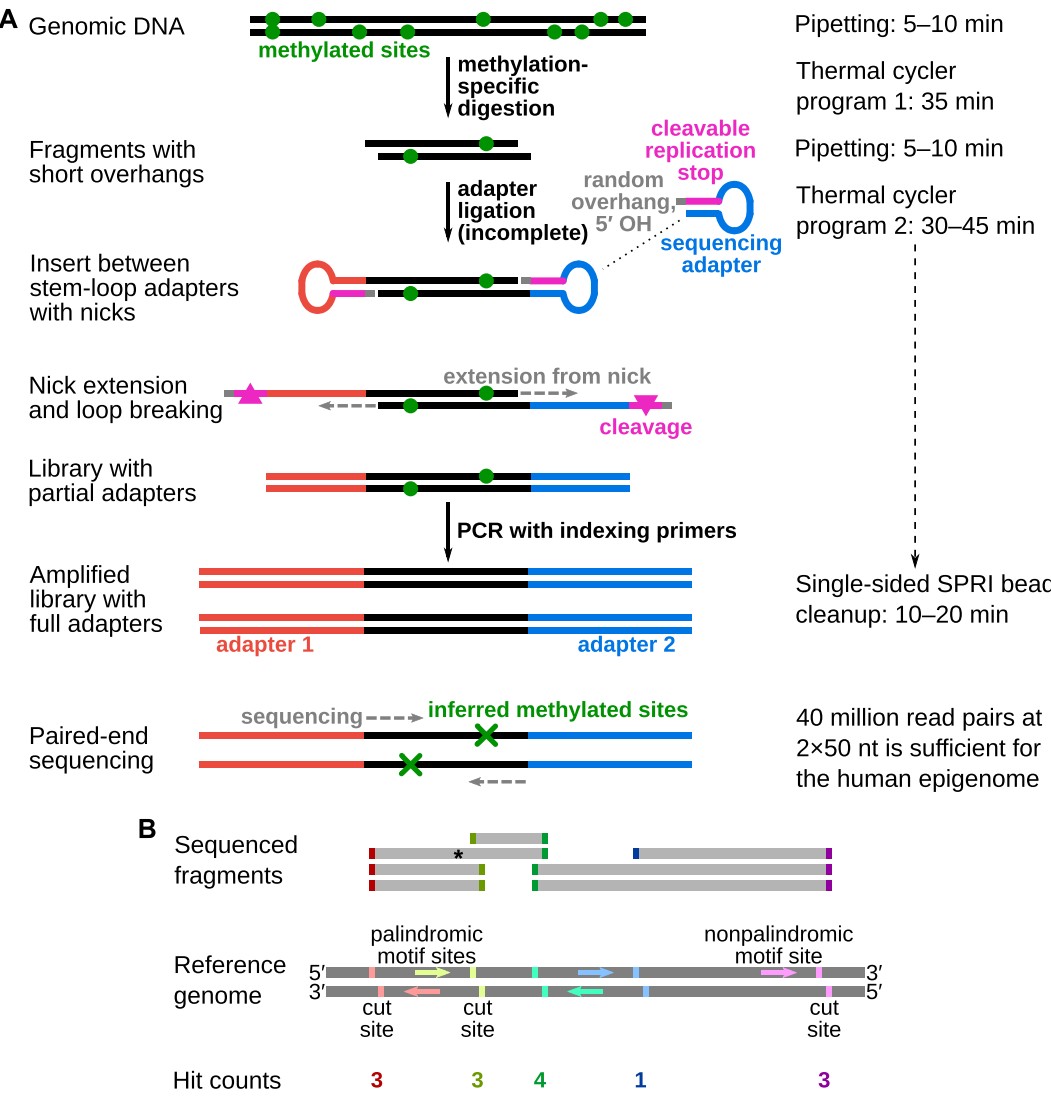

**Figure 1. Diagram of fragmentation at methylated loci and sequencing.**
**(A)** Library preparation reactions. Genomic DNA is digested by a methylation-dependent restriction endonuclease that cuts at a known distance from the methylated cytosine in its motif and leaves a short single-strand overhang of unknown bases. Stem-loop (hairpin) sequencing adapters with complementary random overhangs are ligated to the digested genomic DNA fragments, but the phosphodiester backbone is completed only on one strand because the adapters lack a 5′ phosphate. The resulting single-strand nick is extended by DNA polymerase to fill in a second strand complementary to the adapter's loop, whereas the unneeded stem strand is degraded. This library of genomic DNA inserts between double-stranded linear short adapters is then amplified by standard polymerase chain reaction with long indexing primers to produce a sequencing-ready library. A standard solid-phase reversible immobilization bead cleanup without size selection is sufficient to purify the library. Paired-end sequencing reads imply the location of the two methylated cytosines resulting in each observed fragment. **(B)** Counting fragmentation at methylated loci and sequencing fragments as hits at methylated motif sites. The restriction endonuclease used here, MspJI, cuts at the motif ^mCNNR. Each copy of this motif on either strand implies a potential cut site at a certain distance past its 3′ end. When paired-end sequence reads are aligned to the reference genome, each end of a sequenced fragment counts as one hit for the corresponding motif site; for example, the fragment marked by an asterisk tallies one hit each for the red and green motif sites. The number of hits for a given motif site corresponds to the fraction of genome copies methylated at that motif's cytosine position.

byproduct from dimerization of adapters that ligate directly to each other without a gDNA insert. FML-seq's library protocol uses a combination of several techniques to prevent this behavior: (1) The adapters are added to the gDNA before digestion. After digestion there is a heat denaturation of the restriction endonuclease, and at this step, adapters that annealed to each other during storage at higher concentration may melt apart and reanneal to the ends of new gDNA fragments instead. (2) The adapters lack a phosphate at the 5′ terminus, which is required for DNA ligase to connect that

terminus to a matching 3′ hydroxyl. Thus, neither strand in an adapter dimer can be ligated, and even if two adapters with complementary overhangs temporarily anneal, 4 bp of hydrogen bonding may not keep them together when the temperature is raised in subsequent steps. (3) Before PCR, the same polymerase is used to fill in the second strand of the adapter by extension from the nick where the adapter's 5′ end is not joined to the gDNA in-sert's 3′ end. Even if an adapter dimer holds together at the PCR polymerase's extension temperature, it will also have a nick on the

opposite strand, so the polymerase would have to jump over a gap in the template strand to fill in the new strand. (4) The 5′ end of the stem loop (hairpin) adapter has the 4N sticky-end overhang and a complementary sequence that forms the double-stranded stem. In addition to PCR suppression, this stem could also cause adapter dimerization by melting and reannealing to a different molecule. However, in the stem sequence, all thymine bases are replaced with uracil, and a dU-intolerant proofreading polymerase is used to prevent the stem sequence from being replicated. (5) Furthermore, the stem sequence's uracils are excised before PCR by uracil–DNA glycosylase (UDG), leaving abasic sites that further deter replication and may be fully destroyed by hydrolysis at PCR denaturation temperature. In this protocol, UDG from a hyperthermophile (*Archaeglobus fulgidus*) is included in the combined ligation/loop-breaking/PCR master mix, because the traditional *Escherichia coli* UDG interferes with the low-temperature ligation and would need to be added in an additional step between ligation and PCR, whereas the hyperthermophilic UDG appears inert at ligation temperature. (6) For the Illumina sequencing platform, the adapters are based on the less traditional but equally well-supported Nextera sequence (Tn5 transposon) rather than standard TruSeq, because that sequence is T-rich on the 5′ strand and therefore this protocol's adapters are U-rich in the portion removed by UDG. (7) These adapter sequences are incomplete and require long PCR primers to extend them to full length with multiplexing indexes. Like the Nextera design, the PCR primers are also incomplete and only partially overlap the ligated adapter sequences. Thus, both an adapter and a PCR primer are required to create a full-length product matching both the flowcell's amplification primers and the sequencing primers, so neither the original adapters nor the PCR primers can form sequenceable byproducts alone.

An additional consequence of destroying the 5′ ends of the original adapters is that the random overhang is also destroyed; even if it contains a mismatch to the complementary gDNA sequence that is tolerated by the ligase, the mismatched base is not part of the final molecule's sequence. The final sequence is derived only from the gDNA and the random overhang does not contribute mismatches at the ends of sequence reads.

Previous methods that used MspJI or another methylation-specific restriction endonuclease focused exclusively on the 32-bp fragment produced by two complete, symmetric digestions of a fully methylated CpG (Fig S1C). This requires a precise size selection such as cutting a band out of an acrylamide electrophoresis gel, and discards about 96% of the library product (see Fig S7). In contrast, FML-seq treats every cut site as informative of a methylated base, even where digestion is incomplete or the sequence motif is not palindromic.

## Specificity of FML-seq

FML-seq's specificity for methylated cytosine depends on the endonuclease. Concordant with previous reports (10), digestion of unmethylated gDNA from lambda bacteriophage yielded no detectable library product, unless greatly overamplified (Fig S3A). Reads from methylated gDNA aligned largely at the sequence motif targeted by Dcm methylase (16) as expected; reads from unmethylated gDNA had little enrichment for the methylase's motif but still aligned at the endonuclease's motif (10) (Fig S3B).

The other factor in FML-seq's specificity is whether other kinds of gDNA fragmentation produce ligatable ends. To fit the adapters, a fragment must have a 5′ overhang of 4 nt with a terminal phosphate, which is produced by the endonuclease but unlikely to result from other kinds of fragmentation. Accordingly, no-endonuclease control libraries using gDNA from either fresh cells or degraded archival tissue showed no detectable library product unless greatly overamplified (Fig S4).

## Biological validation

To compare FML-seq with other methods, we prepared libraries from four well-studied human cell lines (Fig S5). The sequence reads from human gDNA had high alignability to the reference genome (Fig S6A). Insert lengths varied widely beyond the canonical 32-bp semipalindromic fragment (Fig S1C) (10), but very few inserts were too short to align (Fig S7). Most reads aligned at the expected sequence motif (Fig S6B) (10).

At a variety of functional genome elements FML-seq signal was unimodal and log-distributed, unlike measurements from WGBS which tended toward the extremes of 0% and 100% methylation, and the greatest dynamic range of FML-seq signal was among gene promoters (Fig S8). Promoters were rich in restriction motif sites despite their short lengths (median 15 sites, 97% with at least 1 site; Fig S9), but promoters with few motif sites had a higher density of extreme FML-seq scores, measured as reads per million per restriction motif (Fig S10). FML-seq signal correlated well with DNA methylation signals detected by other methods (Fig 2A and Table S1) (17, 18): the two-channel base-conversion methods (EPIC, WGBS, RRBS) were most similar to one another, but FML-seq was more similar to them than another one-channel method (MeDIP-seq). FML-seq showed lower methylation than other methods for HeLa-S3 in particular, as widespread hypermethylation diluted the one-channel signal, though such extreme genome dysregulation may not be observed in many experiments. FML-seq's relative signal per promoter (Fig 2B) and estimated fold change (Fig 2C) were correlated with the most comprehensive method, WGBS to a similar degree as the EPIC microarray was correlated with WGBS, though FML-seq's signal included more zeroes. Testing for differential methylation between biological conditions, the list of differentially methylated promoters detected by FML-seq showed concordance with WGBS and greater sensitivity than RRBS (Fig 2D). These results at the level of functional genome elements stood in contrast to analysis at the level of individual cytosine positions, where sequence read counts were too low (65% of single-position counts were zero) to correspond with measurements from base-conversion methods (Fig S12).

## Minimal sequencing conditions

Serial dilutions from 60 ng gDNA input showed successful but lower-quality results down to 6 ng (Fig S14). Because FML-seq analysis counts entire sequence reads rather than bases within each read, long reads are no more useful than short reads as long as they can be confidently aligned to the reference genome; the shortest available read lengths proved sufficient (Fig S13). Given the diminishing returns of additional sequencing depth (Fig S14), sufficient sequencing for a human gDNA sample is roughly

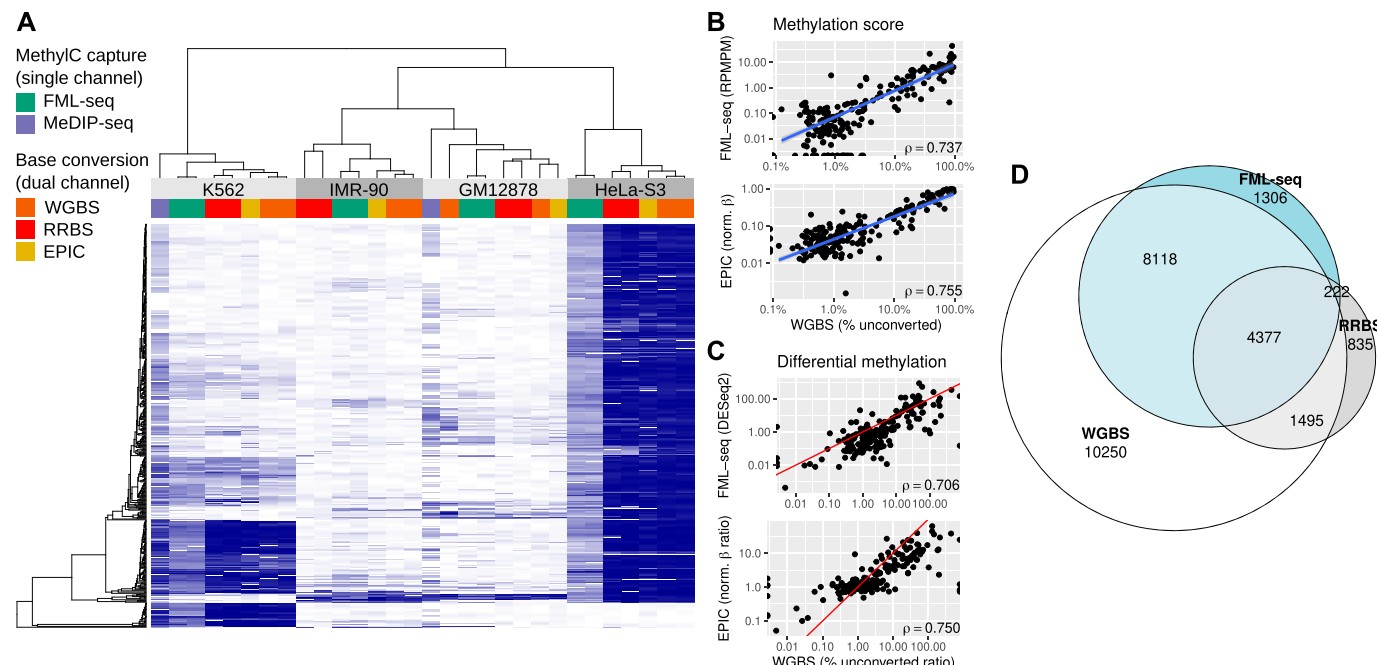

**Figure 2. Fragmentation at methylated loci and sequencing (FML-seq) recovers a similar biological signal as more costly DNA methylation profiling methods on the same samples.**
Methylated DNA immunoprecipitation data were not available for two cell lines. **(A)** Normalized methylation signal (blue: EPIC β, whole-genome bisulfite sequencing [WGBS] and reduced-representation bisulfite sequencing percent methylation, Methylated DNA immunoprecipitation VSD, FML-seq log reads per million per motif) at the 500 promoters with the most variation among cell lines according to WGBS. Technical replicates are shown separately. **(B)** Methylation scores by promoter for K562 in FML-seq or EPIC versus WGBS, replicates pooled. Only 200 randomly sampled promoters are graphed and used for the blue least-squares regression line but Spearman's ρ is calculated with all promoters. See Fig S11 for a visualization of all promoters. **(C)** Methylation fold change by promoter for K562 versus HeLa-S3 in FML-seq or EPIC versus WGBS, replicates pooled. **(B)** The same subset of promoters are graphed as in (B). The red line shows $y = x$. **(D)** Overlap of significantly (BH-adjusted $P < 0.01$) differentially methylated promoters between K562 and HeLa-S3 according to WGBS (24,240 significant), reduced-representation bisulfite sequencing (6,929), and FML-seq (14,023).

40 million read pairs or 96 libraries per NovaSeq S2 flow cell. Thus, although sequencing is the main cost of the FML-seq workflow at current prices, that expense is also more economical than other approaches.

## Discussion

By profiling genome-wide DNA methylation more cost-effectively than current methods, FML-seq will allow new studies with a larger scale of samples than financially feasible with previous methods, whereas the short protocol in 96-well plates will significantly reduce labor and risks of error (Tables 1 and S2). Recently, new reductions in sequencing prices have been forecast to bring the "$100 genome" (19); as sequencing makes up nearly all the cost of FML-seq, cheaper sequencing could also bring the $20 epigenome. Whereas somatic genome sequencing might be informative only once per individual, inexpensive epigenome profiling by FML-seq could be used repeatedly to track diverse variables such as disease progression, tissue differentiation or genomic response to the environment. With FML-seq, epigenetic profiling could become a routine high-throughput assay like qPCR rather than a costly exploratory experiment limited to precious samples.

Whereas dual-channel signals like base conversion have roughly equal sensitivity at all loci they measure, FML-seq's single-channel signal is more precise when the signal is high and less certain when the signal is low. In particular, the FML-seq read scores contain more zeroes than observed from base-conversion sequencing methods. These drop-outs could represent loci with low methylation but also ascertainment biases such as loci with few copies of the restriction motif, loci with low accessibility to sequencing (e.g., unmappable duplicated sequences) or low sequencing depth of the library overall. This greater uncertainty at lower-signal sites is analogous to the heteroskedastic RNA-seq signal from both lower-abundance and shorter transcripts, whose ascertainment bias similarly varies from one transcript to the next but can be assumed equal for a given transcript across different samples, and the accuracy of both FML-seq's methylation scores and its differential methylation ratios appears to be similarly maintained by normalizations. In typical experiments, the goal is not to quantify the absolute methylation state of each locus but rather to identify loci whose methylation differs between biological conditions of interest, just as RNA-seq typically searches for differentially expressed transcripts rather than for the absolute molecular copy number of each transcript. If the loci under examination are individual cytosine positions in the genome, base-conversion methods provide precise quantification but FML-seq measures too few positions at realistic sequencing depth; however, a biological question about DNA methylation is likely to consider entire genome regions, such as

**Table 1. Fragmentation at methylated loci and sequencing require less labor and reagent cost than representative cytosine–methylation profiling methods.**

| | Microarray (Illumina) | WGBS (IDT) | EM-seq (NEB) | RRBS (Zymo) | Targeted BS (Illumina) | FML-seq |
|---|---|---|---|---|---|---|
| Reagent cost | $265 | $90 | $40 | $45 | $280 | $5 |
| Protocol time | 4 d | 1 d | 2 d | 2 d | 2 d | 2 h |
| Limiting scale | Eight-sample chip | 24-tube centrifuge | 96-well plate | 24-tube centrifuge | 24-tube centrifuge | 96-well plate |
| Special equipment | Incubators and water circulator | Sonicator | Sonicator | | Sonicator | |
| Minimum gDNA | 250 ng | 100 pg | 10 ng | 10 ng | 500 ng | 6 ng (1,000 cells) |
| Sequencing reads | | 2 × 150 nt × 375 M | 2 × 150 nt × 375 M | 2 × 50 nt × 90 M | 2 × 100 nt × 55 M | 2 × 50 nt × 40 M |
| Sequencing cost | | $615 | $615 | $180 | $135 | $80 |
| Targeted CpG sites | 0.85 M | 68 M | 68 M | 5 M | 3 M | 36 M |

Cost per sample includes all reagents but not standard consumables (tubes, pipet tips) and is rounded to the nearest five USD. Protocol time includes all steps from isolated gDNA to sequencing-ready library. "Special equipment" for sample preparation excludes common instruments such as pipets, thermal cyclers, centrifuges, and separation magnets. Sequencing read lengths are per the manufacturers' recommendations. Base-conversion sequencing methods use 30X coverage per ENCODE guidelines, assuming 80% read alignment to the human genome. Sequencing costs use list prices for appropriate Illumina NovaSeq 6000 High Output reagents with up to 96-plex indexing. CpG sites are counted in the CHM13v2 reference sequence, both strands.

gene bodies or regulatory elements. Here, we have shown that FML-seq quantitatively detects differentially methylated genome regions.

Because it counts digestions by a methylation-specific endonuclease, FML-seq is in a sense both an enrichment method and a reduced-representation method. In this study, we used MspJI endonuclease as its degenerate recognition motif $^{m}$CNNR is very common. We cautiously restricted our data analysis to $^{m}$CGNR as methylation in the human genome typically occurs at the CpG dinucleotide, but the lambda bacteriophage experiment showed that FML-seq is equally compatible with other genomes in which 5 mC occurs in other contexts. Other endonucleases related to MspJI produce the same 5′ 4-base overhang compatible with FML-seq's adapter design (10), and LpnPI ($^{m}$CDG, which in CpG context is $^{m}$CGG as in RRBS) in particular has also been demonstrated for genome-wide DNA methylation profiling (12); substituting an enzyme with a less common motif would result in a more reduced representation, that is, concentrate more sequence reads at a smaller number of digestion sites, though an additional tradeoff is a reduction in total library yield and therefore an increase in required input DNA because fewer sequenceable fragments are produced by less promiscuous digestion (data not shown). Conversely, if additional methylation-dependent restriction endonucleases become commercially available, a multiple digestion could increase the number of sites measurable by FML-seq. Thus, in addition to its advantages in cost and scalability, FML-seq is a robust modular technique that may be expanded in the future.

# Materials and Methods

### Data generation

#### Reference genomic DNA preparation
HeLa-S3, IMR-90, and K562 gDNAs were purchased from MilliporeSigma (catalog #87110901, 85020204, 89121407). GM12878 gDNA was purchased from the Coriell Institute for Medical Research (#NA12878). Methylated lambda bacteriophage gDNA, from strain cI857 Sam7 grown in *E. coli* strain W3110, was purchased from Thermo Fisher Scientific (#SD0011). Unmethylated lambda gDNA, from the same virus strain grown in *dam⁻ dcm⁻ E. coli* strain GM2163, was purchased from Thermo Fisher Scientific (#SD0021). All gDNA samples were requantified with a Qubit 1X dsDNA High-Sensitivity kit (#Q33230; Thermo Fisher Scientific) and diluted according to this measurement in nuclease-free 1X TE buffer with 0.05% wt/vol Tween 20. All experiments used 60 ng gDNA except those comparing lower inputs, for which gDNA was serially diluted and technical replicates were taken from the same dilution.

#### FML-seq library preparation
Sequencing libraries were prepared according to the FML-seq protocol (Supplemental Data 1) with final holds at 14°C. In the specificity validation experiments, technical replicates were performed side-by-side with different numbers of PCR cycles to detect widely different yields; in these and the experiments comparing different amounts of gDNA, at the end of each measured cycle number before the last, the appropriate tubes were transferred to a dry-block incubator at 72°C for 1 min and then to a 4°C refrigerator to replace the final extension step in the PCR program. In the no-endonuclease control experiment, MspJI restriction endonuclease was replaced by an equal volume of its storage buffer (#B8002S; New England Biolabs).

#### Library quality control and quantification
Libraries from all experiments were profiled, undiluted, on an Agilent TapeStation 4200 with the High Sensitivity D1000 Reagents (#5067-5585). Libraries to be sequenced were diluted to 1/10,000 concentration by serial 1/100 dilutions and quantified by qPCR with the primers 5′-AATGATACGGCGACCACCGA-3′ and 5′-CAAGCAGAA-GACGGCATACGA-3′ at 450 nM each, using PowerUp SYBR Green

Master Mix (#A25742; Thermo Fisher Scientific) and a program of 2 min at 50°C, 2 min at 95°C, then 40 cycles of 15 s at 95°C, 1 min at 60°C. The KAPA Library Quantification DNA Standards #1–5 (#07960387001; Roche) were used for the standard curve. The qPCR measurements were scaled by the qPCR's dilution factor and the TapeStation's average library molecule lengths to calculate the library molarities used for pooling.

### Sequencing

Shallow sequencing with long reads, for initial characterization of the libraries from the methylated versus unmethylated lambda gDNA experiment and the human cell-line gDNA experiment, was performed on an Illumina MiSeq with the MiSeq Reagent Nano v2 kit (#MS-103-1001) yielding paired-end reads of 2 × 154 nt and index reads of 2 × 8 nt, and all libraries from each experiment loaded at equal molarity. Deeper sequencing with short reads was performed for the 60 ng libraries from the human cell-line experiment; these were deliberately sequenced to unnecessary depths to enable rarefaction analysis (see Supplemental Data 1 for sequencing recommendations) using an Illumina NovaSeq 6000 with the S1 Reagent Kit v1.5 (#20028319) yielding paired-end reads of 2 × 61 nt and index reads of 2 × 8 nt. The 6 ng and 600 pg libraries were sequenced on an Illumina NextSeq 500 with the High Output Kit v2.5 (#20024906) yielding paired-end reads of 2 × 38 nt and index reads of 2 × 8 nt.

### Data processing

#### Functional genome element annotations

The NCBI RefSeq functional regulatory element list (20) and the comprehensive set of nonoverlapping "promoter-like" candidate cis-regulatory elements in the human genome (21), totaling 40,891 promoters and 1,450 enhancers, were converted from the hg38 reference genome to T2T-CHM13v2.0 (hs1) (22) by UCSC liftOver (23), eliminating genome elements whose old coordinates were deleted in the new reference. Only candidate cis-regulatory elements with lengths of 150–350 bp in the new reference were kept, matching the range in the old reference. The NCBI RefSeq-curated gene annotations (20) were downloaded in original T2T-CHM13v2.0 coordinates. This yielded final sets of 286,435 distinct exons, 297,593 introns, 40,351 promoters, and 1,402 enhancers. Each sequence motif or microarray probe was counted within a genome element if the position of the cytosine base being tested for methylation state was within the boundaries of the element, even if the rest of the motif or probe sequence lay beyond the boundaries.

#### FML-seq

Standard Nextera adapter sequences were trimmed from all FML-seq reads by cutadapt version 4.1 (24) and the NextSeq trimming option was enabled for NextSeq data. Trimmed reads from human samples were aligned to the T2T-CHM13v2.0 (hs1) reference genome (22), and from lambda bacteriophage samples to the circularized lambda reference genome (RefSeq NC 001416), by bwa-mem2 version 2.2.1 (25). An FML-seq read was counted as a hit at a CGNR site if the beginning of the read (the end of the fragment) aligned at the expected distance of 10 bp from the motif (to the right of a motif on the forward strand or to the left on the reverse strand).

The hit count for each promoter was calculated as the total of all CGNR positions whose first C base was within the promoter. For exploratory analysis, each promoter's hit count was normalized both by the library size and by the number of CGNR motifs within the promoter to reads per million per motif (RPMPM): for any promoter $i$ with hit count $c_i$ and motif count $m_i$,

$$\mathrm{RPMPM}_i = \frac{10^6 c_i}{m_i \Sigma_j c_j}$$

and this value was added to 1 before computing the logarithm, $\log_{10}$ ($\mathrm{RPMPM}_i$ + 1).

### MeDIP-seq

Standard TruSeq adapter sequences were trimmed from all MeDIP-seq reads by cutadapt version 4.1 (24), then the trimmed reads from human samples were aligned to the T2T-CHM13v2.0 (hs1) reference genome (22) by bwa-mem2 version 2.2.1 (25). Each DNA fragment inferred from the alignments of paired-end reads was counted as a hit in a given promoter if the center position of the fragment was within the boundaries of the promoter. The matrix of promoters × samples was processed in DESeq2 version 1.36 (26) with no design variable, as the experiments were unreplicated, and normalized with the variance-stabilizing transformation.

### Bisulfite sequencing

Reads from WGBS and RRBS were processed with similar pipelines. Standard TruSeq adapter sequences were trimmed from all reads by cutadapt version 4.1 (24), then the trimmed reads were aligned to the converted T2T-CHM13v2.0 (hs1) reference genome (22) by bwa-meth version 0.2.5 using bwa-mem2 version 2.2.1 (25). For RRBS reads, the two bases before each adapter were also trimmed as these are not subject to bisulfite conversion, and the minimum score was set to zero in bwa-mem2 because of the short reads (36 nt). Converted and unconverted bases were counted at each CpG position by MethylDackel version 0.6.1 and the methylation score of each promoter was calculated as the total proportion of methylated bases detected at all CpG positions within the promoter, equivalent to the mean proportion of methylation across positions weighted by the coverage at each position.

### Microarray

Genome coordinates of CpG sites targeted by microarray probes were converted from the hg38 reference genome to T2T-CHM13v2.0 (hs1) (22) by UCSC liftOver (23), eliminating sites whose old coordinates were deleted in the new reference or whose base sequence was no longer CG in the new reference. The previous samples from ENCODE were imported in one batch by ChAMP version 2.26.0 (27). The methylation score of each promoter was calculated as the mean $\beta$ value of all probes measuring a cytosine position within the promoter.

### Method clustering

A subset of 9,876 promoters were measurable by every method (at least one EPIC probe, WGBS/RRBS CpG read, MeDIP fragment, or 60 ng FML-seq CGNR read). From this subset, the 500 promoters with the greatest variation among cell lines were selected by the

greatest $\chi^2$ scores from the two-way table of cell type × methylated versus unmethylated hit counts from WGBS with replicates pooled. Each method's 500 promoters × samples matrix of methylation scores (EPIC $\beta$, WGBS, and RRBS percent methylation, MeDIP VSD, FML-seq log RPMPM) was linearly scaled to the range (0, 1) before the matrices were concatenated. Hierarchical clustering was performed by UPGMA on Pearson distances $(1 - r)$ of the concatenated matrix.

## Data Availability

All raw sequencing data generated in this study have been submitted to the NCBI Sequence Read Archive (SRA; https://www.ncbi.nlm.nih.gov/sra) under accession number PRJNA914781. Data-processing pipeline scripts are collected at https://github.com/jwfoley/FMLtools. Scripts used to perform the analyses in this study are collected in Supplemental Data 2.

## Supplementary Information

## Acknowledgements

We are grateful to the laboratory of William D Greenleaf for use of the TapeStation, to Lewis A Marshall and Philippe Jolivet for their invaluable discussion, and to Fumiaki Katagiri and Andrew Z Fire for helpful comments on the article. NovaSeq sequencing was performed by the Stanford Genomics Service Center on an instrument purchased with NIH S10 Shared Instrumentation Grant 1S10OD02521201. Previous data were generated for the ENCODE Consortium by the laboratories of Richard M Myers, Michael P Snyder, and Bradley E Bernstein. The authors were supported by the National Cancer Institute of the National Institutes of Health under awards U2C CA-17-035 and R01CA193694 and by the Breast Cancer Research Foundation under award PPI-18-006.

### Author Contributions

JW Foley: conceptualization, resources, data curation, software, formal analysis, validation, investigation, visualization, methodology, project administration, and writing—original draft, review, and editing.
SX Zhu: investigation.
RB West: conceptualization, supervision, funding acquisition, project administration, and writing—original draft, review, and editing.

### Conflict of Interest Statement

JW Foley is the inventor on a patent covering part of the library synthesis protocol used in FML-seq and involved in forming a company that will commercialize it.

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
