## [Reviewer comments · Life Science Alliance]

Life Science Alliance

Cost-effective DNA methylation profiling by FML-seq

Joseph Foley, Shirley Zhu, and Robert West
DOI: <https://doi.org/10.26508/lsa.202302326>

Corresponding author(s): Joseph Foley, National University of Singapore

Review Timeline:

Submission Date:	2023-08-17
Editorial Decision:	2023-08-23
Revision Received:	2023-08-24
Editorial Decision:	2023-08-25
Revision Received:	2023-09-20
Accepted:	2023-09-21

Transaction Report:

Please note that the manuscript was previously reviewed at another journal and the reports were taken into account in the decision-making process at *Life Science Alliance*. Since the original reviews are not subject to Life Science Alliance's transparent review process policy, the reports and author response cannot be published.

August 23, 2023

Re: Life Science Alliance manuscript #LSA-2023-02326-T

Joseph William Foley
Stanford University School of Medicine

Dear Dr. Foley,

Thank you for submitting your manuscript entitled "Cost-effective DNA methylation profiling by FML-seq" to Life Science Alliance. The manuscript was assessed by expert reviewers at another journal and then transferred to Life Science Alliance. The reviewer comments were assessed at LSA, and LSA editors deemed that the manuscript could be published at LSA provided the authors revise the manuscript to discuss the limitations outlined by Reviewer 2.

Thank you for this interesting contribution to Life Science Alliance. We are looking forward to receiving your revised manuscript.

Sincerely,

B. MANUSCRIPT ORGANIZATION AND FORMATTING:

August 25, 2023

RE: Life Science Alliance Manuscript #LSA-2023-02326-TR

Joseph William Foley
Stanford University School of Medicine

Dear Dr. Foley,

Thank you for submitting your revised manuscript entitled "Cost-effective DNA methylation profiling by FML-seq". We would be happy to publish your paper in Life Science Alliance pending final revisions necessary to meet our formatting guidelines.

- please upload your main manuscript text as an editable doc file
- please upload your Tables in editable .doc or excel format;
- please upload all figure files as individual ones, including the supplementary figure files; all figure legends should only appear in the main manuscript file
- please add a Category for your manuscript in our system
- please add the Twitter handle of your host institute/organization as well as your own or/and one of the authors in our system
- please add your main, supplementary figure, and table legends to the main manuscript text after the references section
- please exclude figures from the manuscript text and upload them separately
- please add authors' contributions to our system
- we encourage you to revise the figure legends for figure S12 such that the figure panels are introduced in an alphabetical order
- please add callouts for Figures S1A-B; S5A-C; S7A-C; S8A-B; S9A-C; S11A-B; S12A-C; S14A-E to your main manuscript text;
- please include the text in the Supplementary file in the main manuscript text

A. FINAL FILES:

B. MANUSCRIPT ORGANIZATION AND FORMATTING:

Sincerely,

September 21, 2023

RE: Life Science Alliance Manuscript #LSA-2023-02326-TRR

Dr. Joseph William Foley
National University of Singapore
Otolaryngology
1E Kent Ridge Road
Singapore 119228

Dear Dr. Foley,

Thank you for submitting your Methods entitled "Cost-effective DNA methylation profiling by FML-seq". It is a pleasure to let you know that your manuscript is now accepted for publication in Life Science Alliance. Congratulations on this interesting work.

DISTRIBUTION OF MATERIALS:

Again, congratulations on a very nice paper. I hope you found the review process to be constructive and are pleased with how the manuscript was handled editorially. We look forward to future exciting submissions from your lab.

Sincerely,
